# Coping strategies and changes in type D personality were associated with depressive tendency at 9 months after percutaneous coronary intervention

**Daisuke Yamaguchi[1], Yoshihiro Asano[1], Koichiro Kuwahara[2], Atsushi Izawa**  [1,2]*

**1** Division of Nursing, School of Health Sciences, Shinshu University, Matsumoto city, Nagano, Japan,
**2** Department of Cardiovascular Medicine, School of Medicine, Shinshu University, Matsumoto city, Nagano, Japan

* izawa611@shinshu-u.ac.jp

**Data Availability Statement:** All relevant data are within the paper and its Supporting Information files.

## Abstract

Type D personality, characterized by negative affectivity and social inhibition, has been associated with both the psychophysiology of coronary artery disease (CAD) and depressive disorders. However, few reports have described the impact of coping strategies in these patients. This study aimed to analyze the characteristics of type D personality and the coping strategies adopted by patients with CAD and to explore the factors associated with depressive tendencies during follow-up. Among 84 patients with CAD (median age 66.5 years, nine women) who underwent percutaneous coronary intervention (PCI), we examined pre-discharge characteristics for personality and coping strategies. We prospectively evaluated associations with the persistence or improvement of depressive tendencies at 9 months. Our findings revealed that persistence of depressive tendencies at 9 months was inversely associated with the adoption of the "planning" coping strategy (odds ratio [OR]: 0.80). We observed worsening depressive tendencies in patients with type D personality who transitioned from non-type D during follow-up. Conversely, improvement in depressive tendencies was associated with the adoption of "planning" (OR: 1.47), "evasive thinking" (OR: 1.47), and "positive interpretation" (OR: 1.43) coping strategies, and inversely associated with the adoption of the "abandonment or resignation" strategy (OR: 0.71). The persistence or improvement of depressive tendencies at 9 months post-PCI was associated with the adoption of specific coping strategies. Changes in type D personality during follow-up were associated with the status of depressive tendency. Personality-oriented treatment incorporating specific coping strategies may provide new strategies to prevent depression and improve care for patients with CAD.

## Introduction

Studies have reported that 15%–60% of patients with coronary artery disease (CAD) have depressive tendencies [1, 2], which is an important risk predictor for CAD [3]. Patients with

**Funding:** D.Y. was funded by Japan Society for the Promotion of Science (JSPS, https://www.jsps.go. jp/english/e-grants/) KAKENHI [grant number JP18K17457]. The funder had no role in the study design, data collection and analysis, decision to publish, or preparation of the manuscript.

**Competing interests:** The authors have declared that no competing interests exist.

CAD and depressive tendencies are less likely to adhere to pharmacotherapy, smoking cessation interventions, and strategies to prevent recurrent cardiovascular events compared to those without depressive tendencies [4, 5]. Psychological care and treatment are necessary for patients with CAD and depression [6]; however, only 4.1% of them undergo antidepressant treatment [7].

In a 12-year longitudinal study involving 1,999 patients aged 55 years and older, time-dependent depressive symptoms, such as depressed mood, insomnia, and loss of appetite, were found to be associated with all-cause and cardiovascular mortality [8]. A meta-analysis of 4,555 patients from nine cohort studies on major adverse cardiovascular events (MACEs) after percutaneous coronary intervention (PCI) revealed that the risk ratio for MACEs in patients with depressive tendencies was 2.10 [9].

The influence of patients' personality traits on MACEs and depressive tendencies has been reported [10]. Type D (distress) personality, characterized by negative affectivity and social inhibition, has emerged as a predictor of CAD onset. Negative affectivity involves the suppression of emotional expression, whereas social inhibition involves suppressing emotional and behavioral expression [10]. Furthermore, type D personality has been identified as a factor associated with depression [11].

The prevalence of type D personality in patients with CAD has been reported as follows: 30% in individuals with angina pectoris, 28% in those with myocardial infarction, and 31% in those with ischemic heart failure [12]. Among healthy individuals, type D personality is associated with depression, with a prevalence ranging from 25% to 31% [13, 14]. The comparable prevalence of type D personality in patients with CAD and healthy individuals suggests that the frequency of this personality trait may not be dependent on the presence of CAD.

We previously investigated 100 patients immediately post-PCI and pre-discharge using the Self-rating Depression Scale (SDS) similar to that used in this study and found that 59.0% had depressive tendencies and 44.0% had a type D personality. A particularly high rate (55.9%) of type D personality was found among individuals with depressive tendencies [15]. In a follow-up survey of 89 post-PCI patients 1 month after discharge, 55.1% showed depressive tendencies and 44.9% had a type D personality. Among patients with depressive tendencies, 59.2% had a type D personality, indicating a high association at 1 month after discharge [16].

The importance of utilizing appropriate coping strategies for stress management has been highlighted [17]. One effective intervention to address depressive tendencies is the implementation of coping strategies consisting of cognitive and behavioral efforts to reduce stress [16]. Research has shown that individuals who do not use effective coping strategies are more likely to experience depressive symptoms [16, 18]. In a comparison of 78 patients with acute coronary syndromes and 146 patients with stable angina pectoris, it was reported that maladaptive coping strategies were associated with acute coronary syndromes, whereas adaptive coping strategies showed a negative association [19]. We investigated coping strategies of pre-discharge CAD patients and found that the adoption of "abandonment or resignation"—a maladaptive coping strategy—had an odds ratio (OR) of 1.33 for depressive tendencies [15]. Conversely, at 1-month post-PCI, the adoption of "planning"—an adaptive coping strategy—was negatively associated with depressive tendencies, with an OR of 0.73 [16].

These findings suggest that stress management using adaptive and approach-oriented coping strategies may be useful in preventing or alleviating depressive tendencies in patients with CAD after hospital discharge [15, 16, 20, 21]. However, few studies have evaluated the relationship between coping strategies, personality characteristics, and changes in depressive tendencies during the chronic phase of CAD. This study aimed to (1) investigate pre-discharge coping strategies and type D personalities in post-PCI patients and (2) evaluate changes in

these factors with time during follow-up to determine their association with the improvement or persistence of depressive tendencies at 9 months after discharge.

## Methods

### Patient and study design

This is a prospective cohort study involving patients with CAD who were hospitalized at Shinshu University Hospital and underwent PCI between July 27, 2016, and June 20, 2017. The study follows the Strengthening the Reporting of Observational Studies in Epidemiology (STROBE) checklist (version 4 for cross-sectional studies) [22]. Patients were excluded if they met any of the following criteria: (i) a history of mental disorder; (ii) dementia; (iii) inability to complete the questionnaire; (iv) disorders of consciousness or severe heart failure; (v) conditions associated with depressive tendencies, such as chronic renal disease [23], cerebrovascular disease [24], malignant neoplasms [25], or respiratory failure [26].

At the time of admission (hereafter, referred to as baseline), self-administered questionnaires were used to collect data on age, sex, employment status (employed/unemployed), living situation (with family/alone), current smoking status, and body mass index. Additionally, cardiovascular risk factors (dyslipidemia, hypertension, diabetes mellitus, peak creatinine kinase, brain natriuretic peptide, number of coronary lesions, number of PCIs, length of hospital stay, and CAD type) were extracted from electronic medical records. Information on underlying diseases, history, and exclusion criteria was carefully reviewed from medical records and physicians' diagnoses.

All participants were informed orally and in writing regarding the nature of the study, their voluntary participation, and their right to withdraw from the study at any time without penalty. Individuals who provided written informed consent were asked to complete the following self-administered questionnaires assessing depressive tendencies, adopted coping strategies, and the presence or absence of type D personality. These questionnaires were administered at baseline and again by mail at 9 months post-PCI when patients were just beyond the high-risk period for target lesion revascularization [27]. The reported prevalence of depression or anxiety in patients with CAD remains constant for up to 12 months [28, 29]. This study aimed to evaluate the psychological impact of PCI during hospitalization in the early chronic phase after discharge. Therefore, we conducted an evaluation at 9 months, referencing studies that reported psychological assessments within 12 months [30–32].

For multivariate logistic regression analysis, the size of the smaller category of the dependent variable should be at least 10 times the number of target independent variables [33]. Given the nature of the statistics and the approximately 50% prevalence of depressive tendencies in patients after PCI, the sample size was set to 80 or more to accommodate up to four independent variables [15, 16]. Eighty-four patients who responded to both the baseline and 9-month questionnaires were included in the analysis (Fig 1). This study was approved by the Medical Ethics Committee of Shinshu University School of Medicine (Approval No. 3428) and was conducted in accordance with the principles of the Declaration of Helsinki.

### Measurements

**Depressive tendency.**  Depressive tendency was assessed using the Japanese version of the SDS [34] developed by Zung [35]. This scale, which has been used to evaluate depressive tendencies in clinical studies of cardiovascular disease [36, 37], employs a four-point Likert scale (ranging from 1 [never or rarely] to 4 [very often]) across 20 questions. A total score of 40 or higher was used to identify depressive tendencies. The SDS has been reported to have significant agreement with physicians' diagnoses, with a high specificity ranging from 72–87%

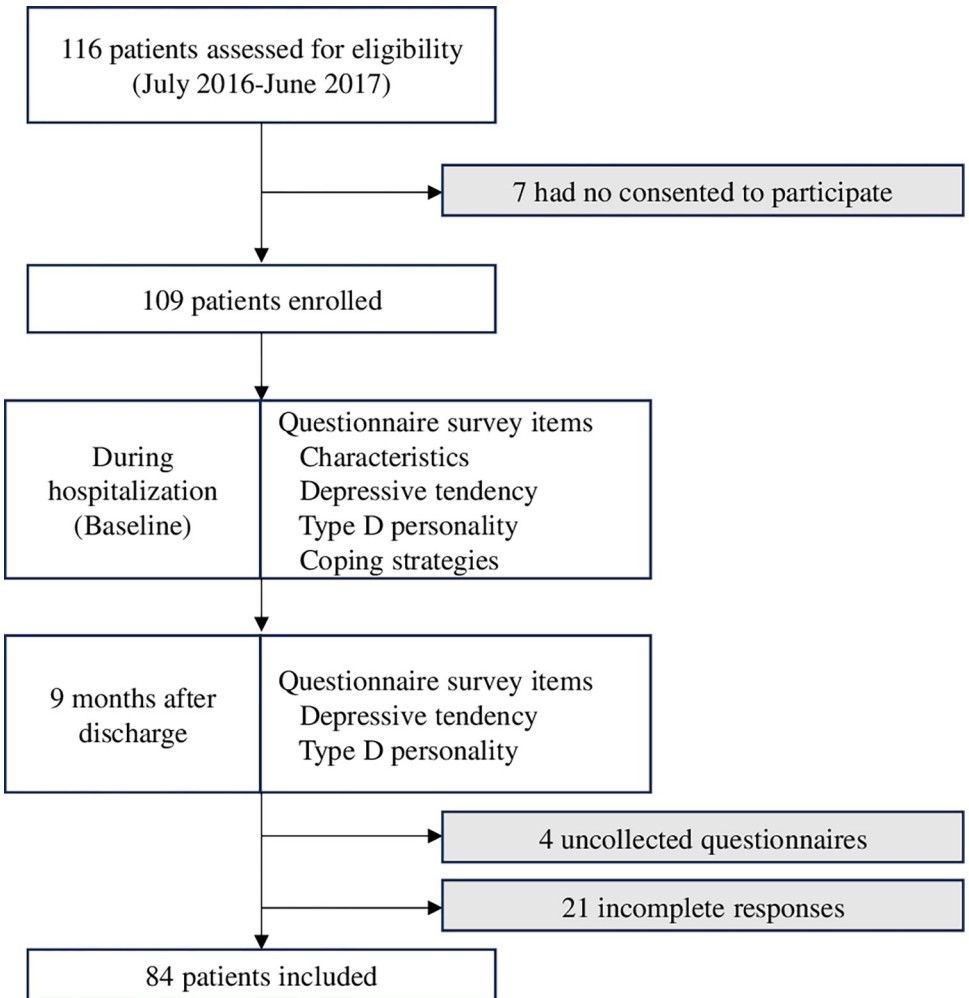

**Fig 1. Survey period and contents.** Of the 116 patients who met the eligibility criteria, 109 signed the consent form. Of the 105 who responded to both questionnaires (at baseline and at 9 months), 84 patients were included in the analysis after excluding 21 patients with missing values.

[38, 39]. In the present study, Cronbach's alpha was 0.748 at baseline and 0.787 at 9 months, consistent with previous reports [34].

**Coping strategies.** Coping strategies were assessed using the Tri-axial Coping Scale (TAC-24), a self-administered questionnaire [40]. This scale evaluates three categories of coping strategies: the problem-focused/emotion-focused axis, the approach-avoidance axis, and the cognitive-behavioral axis [41]. This scale uses a five-point Likert scale (1 [never] to 5 [always]) across 24 questions, with eight subscales to classify coping strategies. The eight types of coping strategies are as follows: (i) *catharsis*, (ii) *abandonment or resignation*, (iii) *information gathering*, (iv) *distraction*, (v) *evasive thinking*, (vi) *positive interpretation*, (vii) *planning*, and (viii) *responsibility shifting*. Based on scores ranging from a minimum of 3 to a maximum of 15 for each type, the coping strategies with the highest scores were identified as the adopted strategy. Cronbach's alpha in this study was 0.802 at baseline, comparable to that of a previous report [40].

**Type D personality.** Type D personality was assessed using the Japanese version of the Type D Personality Scale (DS14), originally developed by Denollet [42, 43]. The DS14 uses a

five-point Likert scale [0 (disagree) to 4 (agree)] for 14 questions, with scores calculated for two subscales: negative affectivity and social inhibition. A type D personality is defined as having a score of 10 or higher on each subscale. Cronbach's alpha for this study was 0.867 at baseline and 0.878 at 9 months, similar to previously reported values [42].

### Statistical analysis

The patients were grouped based on the presence or absence of depressive tendencies at 9 months. Baseline characteristics were compared using the Mann–Whitney U test for continuous variables and the $\chi^2$ test for nominal variables. For comparisons involving three or more variables, Bonferroni correction was applied to account for multiple comparisons. Continuous variables are described as medians [four-part range] because the Shapiro–Wilk test indicated that the variables were not normally distributed.

To evaluate the impact of personality changes on depressive tendency (total SDS score), patients were divided into four groups based on the presence or absence of type D personality and its changes during follow-up. Total SDS scores at baseline were compared with those at 9 months for each group.

Next, the persistence or improvement of depressive tendencies and coping strategies (TAC-24 subscale scores) adopted at baseline were compared using the Mann–Whitney U test. Logistic regression models were then constructed to analyze the association between each coping strategy and the presence or improvement of depressive tendency at 9 months. Separate models were adjusted for age, sex, type D personality, and depressive tendency at baseline [19]. The study included 18 patients with improved depressive tendencies at 9 months and 29 with continued depressive tendencies. Due to the small sample size, logistic regression analyses were exploratory.

All analyses were performed using IBM SPSS Statistics software version 25.0 (IBM. Armonk. USA). A $p$-value <0.05 was considered statistically significant.

## Results

### Patients

Of the 116 patients who were provided with an explanation of the study, 109 signed the consent form; the other seven individuals chose not to participate for unknown reasons. Among the 109 participants, 105 responded to both the baseline and 9-month surveys (response rate: 96.3%). However, 21 respondents with missing values were excluded, resulting in a final analysis cohort of 84 patients (valid response rate: 77.1%) (Fig 1). We did not use other statistical techniques for handling missing values, such as data imputation methods, because imputing missing values could introduce bias depending on the method used, potentially affecting the validity of the results. Furthermore, there were no significant differences in the baseline characteristics between the 84 patients with complete values and patients with missing values.

The baseline characteristics of participants were as follows: median age of 66.5 years, 10.7% female, 60.7% with hypertension, 73.8% with dyslipidemia, 32.1% with diabetes mellitus, and a median body mass index of 24.3 (Table 1 and S1 Table).

### Baseline characteristics and coping strategies associated with depressive tendency at 9 months

At the 9-month evaluation, the prevalence of depressive tendency and type D personality was 47.6% and 40.5%, respectively. The baseline characteristics of 40 patients with depressive tendency at 9 months were compared to 44 patients without depressive tendency (Table 2 and

Table 1. Baseline characteristics of patients (n = 84).

| Characteristics | Median [interquartile range] or n (%) |
|---|---|
| Age (years) | 66.5 [58.0–73.0] |
| Male / female | 75 (89.3) / 9 (10.7) |
| Living with family / alone | 77 (91.7) / 7 (8.3) |
| Employed / unemployed | 52 (61.9) / 32 (38.1) |
| Type D personality | 34 (40.5) |
| Smoking | 21 (25.0) |
| Acute myocardial infarction | 29 (34.5) |
| Hypertension | 51 (60.7) |
| Dyslipidemia | 62 (73.8) |
| Diabetes mellitus | 27 (32.1) |
| Peak creatinine kinase (U/L) | 182 [92–1,402] |
| Brain natriuretic peptide (pg/mL) | 43.0 [22.6–132.0] |
| Coronary vessel involvement | |
| Single-vessel disease | 54 (64.3) |
| Double-vessel disease | 23 (27.4) |
| Triple-vessel disease | 7 (8.3) |
| No. of percutaneous coronary intervention | |
| 1st | 58 (69.0) |
| 2nd | 20 (23.8) |
| 3rd | 6 (7.1) |
| Length of hospital stay (days) | 6 [4–12] |
| Body mass index (kg/m$^2$) | 24.3 [21.7–27.1] |
| Depressive tendency | 47 (56.0) |

S2 Table). This comparison revealed that a significantly higher proportion of patients with depressive tendencies at 9 months were also depressed at baseline (72.5% vs. 40.9%, $p = 0.004$) compared to those without. Additionally, these patients were more likely to adopt the "abandonment or resignation" coping strategy ($p = 0.033$) and less likely to adopt the "planning" and "positive interpretation" strategies ($p = 0.007$ and $0.018$, respectively) compared to those patients without depressive tendencies. These results suggest that baseline depressive tendencies and coping strategies may be associated with depressive tendencies at 9 months.

Based on these findings, we analyzed the ORs of coping strategies for the presence of depressive tendency at 9 months. We found that the adoption of the "abandonment or resignation" strategy was significantly associated with an OR of 1.23 (95% confidence interval [CI], 1.01–1.49). Meanwhile, the adoption of the "planning" strategy was significantly negatively associated with an OR of 0.79 (95% CI, 0.66–0.94), indicating a significant negative correlation. No association was observed between depressive tendencies at 9 months and baseline type D personality with an OR of 1.43 (95% CI, 0.60–3.43). In a multivariate analysis adjusted for baseline age, sex, and depressive tendency, only the "planning" strategy was significant, with an OR of 0.80 (95% CI, 0.66–0.97) (Table 3).

## Changes in type D personality associated with depressive tendency at 9 months

Baseline personality traits were not associated with depressive tendency at 9 months. However, as personality may change over time, we analyzed changes in type D personality and SDS

**Table 2. Baseline characteristics and coping strategies of patients having depressive and those without depressive tendencies at 9 months.**

| | Depressive Tendency (n = 40) | Non-depressive Tendency (n = 44) | p |
|---|---|---|---|
| Baseline characteristics | | | |
| Age (years) | 67.0 [58.0–73.0] | 66.0 [58.0–72.3] | 0.677 |
| Male / female | 39 (97.5)/1 (2.5) | 36 (81.8)/ 8 (18.2) | 0.020 |
| Living with family / alone | 38 (95.0)/2 (5.0) | 39 (88.6)/5 (11.4) | 0.292 |
| Employed / unemployed | 21 (52.5)/19 (47.5) | 31 (70.5)/13 (29.5) | 0.091 |
| Type D personality | 18 (45.0) | 16 (36.4) | 0.421 |
| Smoking | 12 (30.0) | 9 (20.4) | 0.313 |
| Acute myocardial infarction | 15 (37.5) | 14 (31.8) | 0.584 |
| Hypertension | 24 (60.0) | 27 (61.4) | 0.898 |
| Dyslipidemia | 31 (77.5) | 31 (70.5) | 0.463 |
| Diabetes mellitus | 13 (32.5) | 14 (31.8) | 0.947 |
| Peak creatinine kinase (U/L) | 208.0 [82.8–2206.0] | 150.5 [95.3–1387.5] | 0.597 |
| Brain natriuretic peptide (pg/mL) | 45.8 [24.1–156.2] | 41.9 [22.5–117.9] | 0.600 |
| Coronary vessel involvement | | | |
| Single-vessel disease | 25 (62.5) | 29 (65.9) | |
| Double-vessel disease | 12 (30.0) | 11 (25.0) | 0.898 |
| Triple-vessel disease | 3 (7.5) | 4 (9.1) | |
| No. of PCI | | | |
| 1st | 27 (67.5) | 31 (70.5) | |
| 2nd | 10 (25.0) | 10 (22.7) | 0.788 |
| 3rd | 3 (7.5) | 3 (6.8) | |
| Length of hospital stay (days) | 7 [4–13] | 6 [4–10] | 0.315 |
| Body mass index (kg/m$^2$) | 24.0 [21.1–25.6] | 24.8 [22.3–27.2] | 0.154 |
| Depressive tendency | 29 (72.5) | 18 (40.9) | 0.004 |
| Subscales of the tri–axial coping scale | | | |
| Catharsis | 8.0 [6.3–10.0] | 9.0 [6.3–12.0] | 0.113 |
| Abandonment or resignation | 8.0 [6.0–9.0] | 6.0 [5.0–9.0] | 0.033 |
| Information gathering | 9.0 [7.0–10.8] | 9.0 [7.3–11.0] | 0.715 |
| Distraction | 8.0 [6.3–10.0] | 9.0 [7.0–10.0] | 0.734 |
| Evasive thinking | 8.0 [7.0–9.0] | 9.0 [7.0–10.0] | 0.120 |
| Positive interpretation | 9.0 [8.0–11.8] | 10.5 [9.0–14.0] | 0.018 |
| Planning | 9.0 [8.0–11.0] | 11.0 [9.0–13.8] | 0.007 |
| Responsibility shifting | 5.0 [4.0–6.0] | 4.0 [3.0–6.0] | 0.303 |

Data are shown as n (%) or median [interquartile range]. PCI: percutaneous coronary intervention.

Significant differences were observed in the following: "Male/female," "Abandonment or Resignation," "Positive interpretation," and "Planning."

scores from baseline to 9 months (Fig 2 and S4 Table). Nine of 50 patients (18%) with non-type D personality at baseline changed to type D personality by 9 months, and their SDS scores increased significantly (36.0 [32.5–47.0] *vs.* 46.0 [36.0–49.0], *p* = 0.025), suggesting a worsening of depressive tendency. Conversely, 12 of 34 patients (35%) who transitioned from type D personality to non-type D personality demonstrated a significant decrease in SDS scores (44.0 [39.0–51.5] *vs.* 35.5 [35.0–42.8], *p* = 0.037). These findings indicate that transitioning to type D personality was associated with worsening depressive tendencies, whereas transitioning away from type D personality was associated with improvement.

**Table 3. Impact of baseline coping strategies on the depressive tendency at 9 months.**

| Subscales of the tri-axial coping scale | Univariable | | Adjusted model 1 | | Adjusted model 2 | |
|---|---|---|---|---|---|---|
| | OR | 95% CI | OR | 95% CI | OR | 95% CI |
| **Abandonment or Resignation** | 1.23 | 1.01–1.49 | 1.17 | 0.95–1.44 | 1.16 | 0.94–1.45 |
| **Planning** | 0.79 | 0.66–0.94 | 0.80 | 0.66–0.97 | 0.80 | 0.66–0.97 |
| **Positive interpretation** | 0.82 | 0.70–0.97 | 0.86 | 0.73–1.02 | 0.86 | 0.72–1.03 |
| **Catharsis** | 0.86 | 0.74–1.01 | | | | |
| **Information gathering** | 0.95 | 0.82–1.11 | | | | |
| **Distraction** | 0.96 | 0.79–1.15 | | | | |
| **Evasive thinking** | 0.88 | 0.73–1.06 | | | | |
| **Responsibility shifting** | 1.14 | 0.87–1.49 | | | | |

Odds ratios (OR) and 95% confidence intervals (CI) for depressive tendencies at 9 months were calculated using each subscale of the tri-axial coping scale. Model 1 was adjusted for age, sex, and depressive tendency at baseline, and model 2 was adjusted for age, sex, depressive tendency, and type D personality at baseline. In both models with adjustment, only "Planning" demonstrated an association with depressive tendencies at 9 months.

## Association of baseline coping strategies with improvement in depressive tendency at 9 months

Of the 47 patients (56%) who demonstrated depressive tendency at baseline, 18 (21.4%) showed improvement, and 29 (34.5%) remained depressed at 9 months (Table 4 and S3 Table). Among patients who improved, approach-oriented coping strategies such as "planning," "positive interpretation," and "evasive thinking," were predominantly used at baseline. Conversely, avoidance-oriented coping strategies, particularly "abandonment or resignation," were less frequently employed. Further logistic regression analyses, adjusted for age, sex, and baseline Type D personality, identified that the adoption of "planning," "evasive thinking," and "positive interpretation" strategies was positively associated with improvement in

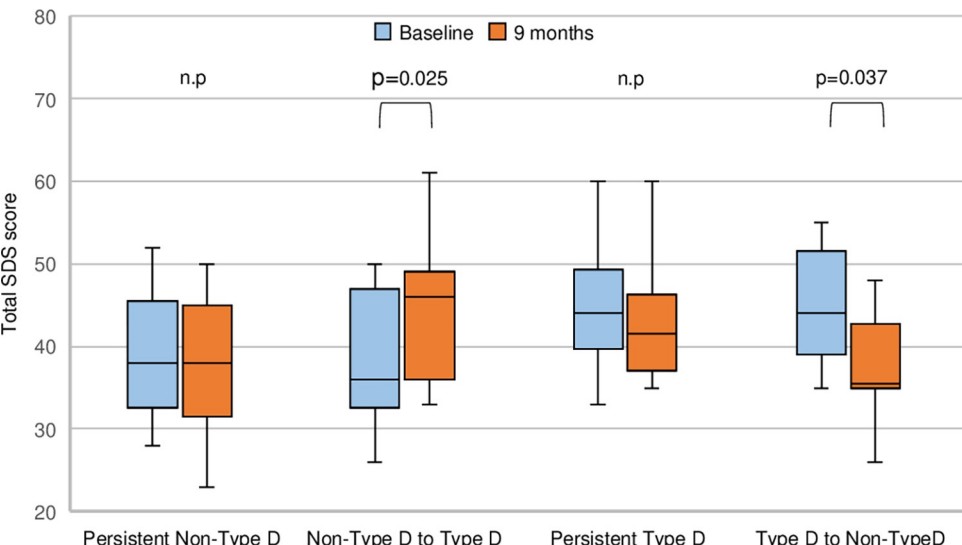

**Fig 2. Total SDS scores in patients with personality changes between type D and non-type D.** The four groups are as follows: a group that remained non-type D, a group that changed from non-type D to type D, a group that remained type D, and a group that changed from type D to non-type D. SDS: self-rating depression scale.

**Table 4. Differences in coping strategies adopted by patients with improvement in and persistent depressive tendency at 9 months compared to the baseline.**

| Subscales of the tri–axial coping scale | Improvement (n = 18) | Persistent (n = 29) | *p* |
|---|---|---|---|
| Catharsis | 8.5 [6.0–12.0] | 8.0 [7.0–10.0] | 0.577 |
| Abandonment or Resignation | 6.0 [5.8–9.0] | 8.0 [7.0–10.0] | 0.013 |
| Information gathering | 9.0 [6.8–10.3] | 8.0 [7.0–10.5] | 0.938 |
| Distraction | 8.5 [7.0–9.3] | 9.0 [7.0–10.0] | 0.991 |
| Evasive thinking | 9.0 [7.8–11.3] | 8.0 [7.0–9.0] | 0.031 |
| Positive interpretation | 12.0 [10.0–13.0] | 9.0 [7.5–10.5] | 0.022 |
| Planning | 11.5 [8.75–14.0] | 9.0 [7.5–10.5] | 0.015 |
| Responsibility shifting | 4.0 [3.0–6.0] | 5.0 [3.5–6.5] | 0.112 |

Data are shown as median [interquartile range].

Comparison of baseline coping strategy subscales between the group with depressive tendency at baseline but not at 9 months and the group with depressive tendency at both baseline and 9 months revealed significant differences in "Abandonment or Resignation," "Evasive thinking," "Positive interpretation," and "Planning."

depressive tendency, whereas the "abandonment or resignation" strategy was negatively associated. These ORs for each coping strategy subscale suggested similar associations with depressive tendency after adjustments (Table 5).

## Discussion

To identify factors associated with depressive tendency in post-PCI patients with CAD, we compared personality traits and coping strategies assessed pre-discharge with those at 9 months and found the following results: (i) Depressive tendency at 9 months was negatively associated with the adoption of the "planning" strategy; (ii) improvement in depressive tendency during follow-up was observed in patients who transitioned from type D to non-type D personality, whereas the progression of depressive tendency was observed in patients who transitioned from non-type D to type D personality; (iii) among the pre-discharge coping strategies, "planning," "evasive thinking," and "positive interpretation" were positively associated with improvement in depressive tendency, whereas "abandonment or resignation" was negatively associated. These findings suggest that the adoption of specific coping strategies

**Table 5. Association of coping strategies at baseline with improvement in depressive tendency at 9 months.**

| Subscales of the tri-axial coping scale | Univariable | | Adjusted model 1 | | model 2 | | model 3 | |
|---|---|---|---|---|---|---|---|---|
| | OR | 95% CI | OR | 95% CI | OR | 95% CI | OR | 95% CI |
| Abandonment or Resignation | 0.72 | 0.52–0.98 | 0.72 | 0.52–0.99 | 0.68 | 0.47–0.97 | 0.71 | 0.50–0.99 |
| Evasive thinking | 1.45 | 1.06–1.89 | 1.50 | 1.08–2.10 | 1.34 | 0.97–1.86 | 1.47 | 1.07–2.02 |
| Positive interpretation | 1.42 | 1.10–1.85 | 1.45 | 1.10–1.89 | 1.37 | 1.05–1.80 | 1.43 | 1.09–1.86 |
| Planning | 1.47 | 1.10–1.95 | 1.50 | 1.11–2.02 | 1.54 | 1.12–2.10 | 1.47 | 1.10–1.96 |
| Catharsis | 1.12 | 0.91–1.38 | | | | | | |
| Information gathering | 1.08 | 0.87–1.33 | | | | | | |
| Distraction | 0.98 | 0.76–1.27 | | | | | | |
| Responsibility shifting | 0.79 | 0.55–1.13 | | | | | | |

Odds ratios (OR) and 95% confidence intervals (CI) for improvement in depressive tendencies were calculated for each subscale of the tri-axial coping scale, adjusted for baseline age in model 1, baseline sex in model 2, and baseline type D personality in model 3. All variables, except model 2 for evasive thinking, were significant. If upper or lower ranges of 95% CI were close to 1.0, the significance of OR and its association should be considered as minimal.

pre-discharge and changes in personality traits over time influence the persistence or improvement of depressive tendency, which may, in turn, affect the prognoses of patients with CAD.

## Associations between depressive symptoms at 9 months and baseline characteristics

The baseline characteristics of participants in the present study were consistent with those of post-PCI patients who had participated in previous large-scale studies [44, 45]. Previous cross-sectional studies have shown that maladaptive coping strategies, such as avoidance-oriented strategies, including "abandonment or resignation," are associated with depression in CAD patients, with ORs of 1.24 (21) or 1.33 [15]. Conversely, adaptive coping strategies, such as "task-oriented" or "planning" have been negatively associated with depression, with ORs of 0.73 [16] or 0.77 [21]. These findings are similar to those confirmed prospectively in the present study. Patients using the "abandonment or resignation" strategy may avoid confronting reality and passively escape from stress, leading to a higher likelihood of depressive tendencies [15].

The effectiveness of stress management programs for patients with CAD, including the adoption of "planning" strategies, has been reported. Incorporating specific "planning" strategies, such as prioritizing and time management, into regular cardiac rehabilitation has resulted in significantly lower levels of chronic depression and stress compared to standard cardiac rehabilitation alone [46]. Additionally, cognitive behavioral therapy [47, 48] and psychotherapy [49] aimed at improving cardiovascular disease outcomes employ "planning" as a coping strategy. For example, cognitive-behavioral therapy that includes "developing plans for daily activities has been shown to significantly reduce scores on the Hamilton depression scale and Hamilton anxiety scale while also improving quality of life and satisfaction [47]. Several reports have explored the relationship between stress management and MACEs. A cohort study of 57,017 patients with no history of cardiovascular disease found that stress management using an "approach-oriented" coping strategy reduced the incidence of stroke and cardiovascular disease mortality [50]. Furthermore, a 6-year prospective cohort study demonstrated that adaptive coping strategies were protective against MACEs, whereas maladaptive coping strategies were associated with an increased risk of MACEs [51]. Conversely, a meta-analysis of 23 randomized controlled trials examining stress management interventions in adults with heart failure found that while short-term interventions improved anxiety, depressive symptoms, disease-specific quality of life, and exercise capacity, the long-term effects remained unclear [52]. Both current and resolved medical conditions, including CAD have been associated with insomnia and impaired well-being [53], life-changing events have been directly linked with sleep disturbances [54]. Therefore, care treatment should also address sleep disorders, which have significant interactions with depressive tendencies and CAD [55, 56]. Further research is needed to develop instructional programs that address both cardiovascular diseases and sleep disorders.

## Coping strategies at the time of hospitalization related to improvement in depressive tendencies at 9 months

In the present study, beside "planning," the coping strategies associated with improvement in depressive tendency at 9 months included two types of passive coping: "evasive thinking," which involves avoiding thoughts about the perceived problem and "positive interpretation," [40] which entails thinking optimistically and affirming positive outcomes. A study evaluating coping strategies in 657 patients with CAD found that 21% of patients adopted passive coping, which was associated with low emotional distress [57]. This is consistent with the findings of

the present study, suggesting that passive coping can be beneficial. Overall, specific coping strategies, both passive and active, appear to be associated with the improvement of depressive tendencies in patients with CAD. Therefore, it is important to provide individualized guidance in selecting coping strategies based on each patient's personality, and the effectiveness of these strategies should be verified. Interestingly, stepwise psychotherapy may benefit depressed patients with CAD with type D personality, whereas such benefits were not observed in patients with non-type D personality [58]. Thus, personality-specific interventions should be considered for patients with CAD. However, due to the small sample size, further research is needed to investigate the individual associations of coping strategy subscales with personality traits and depressive tendencies.

### Depressive tendencies at 9 months and the influence of type D personality during hospitalization

Previous studies have reported an association between type D personality and chronic depressive tendencies in patients with CAD post-discharge [11, 59]. However, in the present study, type D personality assessed pre-discharge [60, 61] was not associated with chronic depressive tendencies. Since personality traits can change due to clinical and life event experiences, the present study evaluated the association between changes in personality traits and depressive tendencies. Results showed that 35% of patients with type D personality who changed to non-type D personality by the 9-month follow-up had a significant decrease in SDS scores. These findings indicate that both changes in personality traits and adopted coping strategy influence the persistence or improvement of depressive tendencies. Currently, there is limited evidence regarding psychological interventions that can change type D personality [62]. However, a recent study has demonstrated that stepwise psychotherapy interventions can reduce the scores of the type D personality scale [58]. Since the mechanisms underlying changes in type D personality were beyond the scope of this study, future research is required to elucidate these mechanisms.

It is important to note that type D personality pre-discharge was not found to be associated with chronic depressive tendency in this study. Conversely, type D personality in patients with CAD has been previously linked with depression [11, 12, 15, 59], and known cardiovascular risks, such as high blood pressure, smoking, a sedentary lifestyle [12]. Future studies should clarify whether type D personality is an independent predictor of unfavorable MACEs, given that personality can change over time and is associated with other cardiovascular risks.

### Limitations

The study has the following limitations. First, there could be a risk of selection bias in the small cohort from our single facility. Second, due to the limited number of variables that could be selected in the multivariate analysis, interactions or independence of different coping strategies could not be evaluated. Third, a small population of women, which was consistent with the sex-dependent distribution of CAD in Japan, limited the analysis for sex-specific differences. Fourth, multivariate analysis was not conducted due to the small sample size for changes in personality traits and coping strategies during follow-up. Fifth, there was no detailed data on dyslipidemia and diabetes. Sixth, self-reporting bias may be present in the design of this study, particularly among patients with depressive tendency who may have reduced motivation or concentration to complete the questionnaire. Seventh, the follow-up evaluation was conducted only at a single time point, 9 months after discharge. Long-term longitudinal studies that account for the potential impacts of other life events on subsequent mental health outcomes after PCI are needed.

## Conclusion

This study demonstrated that the pre-discharge adoption of stress coping strategies and changes in type D personality were associated with the persistence or improvement of depressive tendencies in patients with CAD. Based on the association between personality and the psychopathology of depressive tendency, personalized feedback interventions targeting type D personality may offer novel approaches for delivering patient-centered care. In conclusion, personality-oriented interventions and/or education incorporating specific coping strategies may be effective in preventing and alleviating depression, which could, in turn, positively impact the prognosis of patients with CAD.

## Supporting information

**S1 Table. The values for baseline characteristics of patients.** All values behind the median and interquartile range and the number of patients shown in Table 1.
(XLSX)

**S2 Table. The values for baseline characteristics and coping strategies of patients having depression and those without depressive tendencies at 9 months.** All values behind the median and interquartile range and the number of patients shown in Table 2.
(XLSX)

**S3 Table. The values for differences in coping strategies adopted by patients with improvement in and persistent depressive tendency.** All values behind the median and interquartile range shown in Table 4.
(XLSX)

**S4 Table. The values for total SDS scores in patients with personality changes.** All values behind the box-and-whisker plots shown in Fig 2.
(XLSX)

## Acknowledgments

We thank all the participants in this study for their cooperation. We also extend our sincere thanks to all staff of the Department of Cardiovascular Medicine at Shinshu University Hospital for their support and assistance.

## Author Contributions

**Conceptualization:** Daisuke Yamaguchi, Atsushi Izawa.

**Formal analysis:** Daisuke Yamaguchi, Yoshihiro Asano, Atsushi Izawa.

**Funding acquisition:** Daisuke Yamaguchi.

**Investigation:** Daisuke Yamaguchi, Koichiro Kuwahara, Atsushi Izawa.

**Project administration:** Daisuke Yamaguchi, Yoshihiro Asano, Koichiro Kuwahara, Atsushi Izawa.

**Writing – original draft:** Daisuke Yamaguchi.

**Writing – review & editing:** Yoshihiro Asano, Atsushi Izawa.

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
