## [Decision Letter · Decision Letter 0]

22 May 2024

PONE-D-24-09289Coping strategies and changes of type D personality were associated with depressive tendency at 9 months after percutaneous coronary interventionPLOS ONE

Dear Dr. Izawa,

Thank you for submitting your manuscript to PLOS ONE. After careful consideration, we feel that it has merit but does not fully meet PLOS ONE’s publication criteria as it currently stands. Therefore, we invite you to submit a revised version of the manuscript that addresses the points raised during the review process.

**ACADEMIC EDITOR:** All issues raised by expert reviewers are required.

We look forward to receiving your revised manuscript.

Kind regards,

Vincenzo Lionetti, M.D., PhD

Academic Editor

PLOS ONE

2. We note that your Data Availability Statement is currently as follows: [All relevant data are within the manuscript.]

Reviewers' comments:

Reviewer's Responses to Questions

**Comments to the Author**

1. Is the manuscript technically sound, and do the data support the conclusions?

Reviewer #1: Yes

Reviewer #2: Yes

2. Has the statistical analysis been performed appropriately and rigorously? 

Reviewer #1: Yes

Reviewer #2: Yes

3. Have the authors made all data underlying the findings in their manuscript fully available?

Reviewer #1: Yes

Reviewer #2: Yes

4. Is the manuscript presented in an intelligible fashion and written in standard English?

Reviewer #1: Yes

Reviewer #2: No

5. Review Comments to the Author

Reviewer #1: Abstract

Please define the origin of Type D personality shortly

Line 38: tendency was observed in patients

Line 40, 42, 44: with the adoption of

Line 44: Changes in type D personality

A closing sentence after the results is missing (Conclusion).

Introduction

The definition of a type D personality is missing. This should be integrated.

Line 53: Which strategies are appropriate? Add some examples.

Line 56: aged 55 years and older

Line 68: What is the meaning of "pathological condition"?

Line 70: The percentages of healthy individuals who have type D personality are quite similar compared to the percentages of clinical samples which you have mentioned in the sentence before. This should be highlighted.

Line 74: Why 55.9 % and not 56 % (summing up to 100 %)?

Line 72ff: Are the assessment tools for depressive tendencies the same or different? Name the questionnaires.

Line 105: Which kind/version of STROBE checklist do use?

The sentences with findings of other studies regarding Odds Ratio should be written less technically and more vivid.

The paragraphs in the introduction part could be linked more smoothly.

Methods - Measurements

Reasons for the questionnaires used should be briefly stated.

Example items could be integrated for a better understanding.

The coping strategies could be executed more.

Figure 1: "Patienst with missing values" are excluded. Why do not use imputation as a statistical method?

Line 150 Coping strategies: Is it a self-administered questionnaire? What does tri-axial mean?

Line 177: You use several statistical methods but the sample size is quite low. Therefore, some calculations could be only exploratory. Please, mention the problem of multiple testing.

Results

Line 189: Are there any sociodemographic or other differences between respondents with complete values and with missing values?

Table 1: Have you informations about the status as past smoker?

Why do you take creatinine kinase? Dyslipidemia is quite general. Do you have further informations? If no, mention this a limitation of the study.

Line 199: "frequency of depressive tendency" Please, clarify in this case and other cases the measurement point (baseline, 9 months later or both?).

Line 204: "abandonment or resignation" For the discussion section: What is coping and what is a symptom of depression? Discuss it.

Subheading – Patients: sometimes patients/participants -> considerate using one term

Sometimes it is about the depressive tendency of participants sometimes that participants are depressive. This is a difference. According to the introduction the study is about depressive tendency. The parts writing about depressive patients/participants should be adapted. This should also be taken into account in table 2: depressive - non-depressive. There is no correction for multiple testing (like Bonferroni correction).

Line 230: Where is the asssessment (statistical data) of this missing correlation?

Discussion

Lie 281: "planning strategy" at baseline or 9 months later. A figure regarding the different questionnaires at different measurement points could be helpful.

No enumeration of outcomes and no reporting of outcomes (naming of OR), these should be removed and the sections reworded.

A major limitation is multiple testing. This should be mentioned in the discussion.

Personality traits are quite stable over the life time. Please, discuss why type D behavior patterns should change within several months (trait vs. state). Furthermore, it is necessary to discuss the limitations of the type D concept.

Some CI (see table 5) are quite near by 1.0 (0,5-0,99) re abandonment or resignation. Discuss this, please.

Line 320: There is the the so-called SPIRR-CAD study on depressed CAD patients with and without type D personality traits with psychodynamic parts of the intervention.

Line 339: the present study

Some practical conclusions could be elaborated more specific.

Figures in poor quality, therefore unreadable. Figure description in different font than text.

Table 4: Why median and not mean?

Table 5: Numbers stand for models, not for literature. Please, clarify it.

Reviewer #2: Introduction

The introduction appears to be a little bit confused with an extensive list of data that leads the reader to get lost in the reading. Then, type D personality and depressive tendency which should be the central constructs of the introduction with coping strategies are explained in a superficial manner (type D personality is explained in less than one sentence in line 64) without making the reader understand precisely what is being talked about.

I suggest restructuring the introduction, starting with a clear and comprehensible explanation of type D personality, depressive tendency and coping strategies (even if only with a general explanation, and then focusing in more detail on the specific coping strategies discussed later on); then continuing with all the literature data concerning the association between these personality 'styles', cardiovascular risk and coping strategies, and then concluding with the aims of the paper.

This will allow a more linear reading of the work.

Materials and methods

At line 107, the fourth point of exclusion criteria, which are the “serious illnesses” that would preclude the participation in the study, and which are the differences between these “serious illnesses” and the conditions associated with depressive tendency stated at the fifth point of exclusion criteria?

At line 126, please explain more deeply the statistic used for the calculation of sample size.

What is the rationale behind the choice of repeat the measurements after 9 months and not, for example, after 12 months?

Statistical Analysis

Lines 172 - 174, when explaining the division into four groups, please indicate which four groups are, for the sake of clarity, to avoid any misunderstanding. I suggest adding, at the end of the statistical analysis in the sentence “then the total SDS scores at baseline was compared with those at 9 months” the words :” for each group”, to better explain what you are doing.

Results

The results section contains a huge list of data in which it is easy to get lost, I recommend at the end of each section to make a small summary of the results obtained, even in tabular form, as the authors prefer, in order to fix the results in a more readable form.

Discussion and Conclusion

I suggest dividing the discussion section into an initial 'introductory' part where there is a general discussion of what happened (line 278-291), then a separated discussion for each aim of the study (first aim: line 292-334; second aim: 335-352).

In lines 304-310 I would add, when talking about psychotherapy to improve coping strategies and to have an improvement in depressive tendency, that it is well known that even when diseases are cured, there are important sequelae in the well-being and sleep quality of formerly ill subjects (10.3390/ijerph21020129 ), the sleep quality factor can have a heavy influence on the maintenance or worsening of depressive tendency (10.1016/j.psychres.2020.113239). Thus, taking improved sleep hygiene into consideration, in addition to interventions aimed at improving the well-being of the subjects, may lead to improved prognosis outcomes. I would suggest including the two important papers I reported to define this aspect regarding sleep hygiene in the discussion and add the prospect of improved 'sleep hygiene in the conclusions as a means to achieve an improvement in depressive tendency and thus a better prognosis.

6. PLOS authors have the option to publish the peer review history of their article (what does this mean?). If published, this will include your full peer review and any attached files.

Reviewer #1: No

Reviewer #2: No

---

## [Author Response · Author response to Decision Letter 0]

22 Aug 2024

Please see our point-by-point responses on the attached file:Response to Reviewers.

We thank the reviewers for thoughtful suggestions and insights, which have enriched the manuscript and produced a better and more balanced account of the research. We hope that the revised manuscript is now suitable for publication in your journal.

---

## [Decision Letter · Decision Letter 1]

6 Sep 2024

PONE-D-24-09289R1Coping strategies and changes of type D personality were associated with depressive tendency at 9 months after percutaneous coronary interventionPLOS ONE

Dear Dr. Izawa,

Thank you for submitting your manuscript to PLOS ONE. After careful consideration, we feel that it has merit but does not fully meet PLOS ONE’s publication criteria as it currently stands. Therefore, we invite you to submit a revised version of the manuscript that addresses the points raised during the review process. Please submit your revised manuscript by Oct 21 2024 11:59PM. If you will need more time than this to complete your revisions, please reply to this message or contact the journal office at plosone@plos.org. Please include the following items when submitting your revised manuscript:A rebuttal letter that responds to each point raised by the academic editor and reviewer(s). You should upload this letter as a separate file labeled 'Response to Reviewers'.A marked-up copy of your manuscript that highlights changes made to the original version. You should upload this as a separate file labeled 'Revised Manuscript with Track Changes'.An unmarked version of your revised paper without tracked changes. You should upload this as a separate file labeled 'Manuscript'.If applicable, we recommend that you deposit your laboratory protocols in protocols.io to enhance the reproducibility of your results. Protocols.io assigns your protocol its own identifier (DOI) so that it can be cited independently in the future. For instructions see: https://journals.plos.org/plosone/s/submission-guidelines#loc-laboratory-protocols. Additionally, PLOS ONE offers an option for publishing peer-reviewed Lab Protocol articles, which describe protocols hosted on protocols.io. Read more information on sharing protocols at https://plos.org/protocols?utm_medium=editorial-email&utm_source=authorletters&utm_campaign=protocols.

We look forward to receiving your revised manuscript.

Kind regards,

Vincenzo Lionetti, M.D., PhD

Academic Editor

PLOS ONE

Journal Requirements:

Additional Editor Comments:

All issues raised by expert reviewer are required

Reviewers' comments:

Reviewer's Responses to Questions

Comments to the Author

1. If the authors have adequately addressed your comments raised in a previous round of review and you feel that this manuscript is now acceptable for publication, you may indicate that here to bypass the “Comments to the Author” section, enter your conflict of interest statement in the “Confidential to Editor” section, and submit your "Accept" recommendation.

Reviewer #1: (No Response)

Reviewer #2: All comments have been addressed

2. Is the manuscript technically sound, and do the data support the conclusions?

Reviewer #1: Yes

Reviewer #2: Yes

3. Has the statistical analysis been performed appropriately and rigorously? 

Reviewer #1: Yes

Reviewer #2: Yes

4. Have the authors made all data underlying the findings in their manuscript fully available?

Reviewer #1: No

Reviewer #2: Yes

5. Is the manuscript presented in an intelligible fashion and written in standard English?

Reviewer #1: Yes

Reviewer #2: Yes

6. Review Comments to the Author

Reviewer #1: Second Review for PONE-D-24-09289

Sample Size Justification:

While it is discussed the rationale for using a sample size of 80 patients, the choice of a 9-month follow-up rather than a more conventional 12-month period is still only briefly explained. Further elaboration on why this timeline was selected and how it may impact the generalizability of the findings would strengthen the study's robustness.

Handling of Missing Data:

Although you provided a table comparing respondents with complete and missing data, there is no clear explanation of how missing data was handled in statistical analysis. It remains unclear whether any statistical techniques were considered and why you chose exclusion over other methods.

Exploration of Limitations:

While you have some discussion of limitations, especially regarding the small sample size, other limitations, such as the potential biases introduced by self-reported measures, have not been fully explored. The study’s reliance on self-reporting for both depression and coping strategies could introduce reporting bias, which should be acknowledged more explicitly.

Concluding Remarks:

The conclusion remains somewhat broad, focusing on the potential for personality-oriented interventions without clearly linking this to the specific findings of the study. Adding more concrete suggestions for clinical practice or future research based on the study’s findings could strengthen the practical implications of the study.

Conclusion:

You have made significant improvements, particularly in the areas of clarity, structure, and transparency. However, some refinements could be made. Overall, this manuscript is much more suitable for publication, but the minor revisions could further enhance its quality and impact.

References: Please, check the SPIRR-CAD trial reference. In your text you describe the study with reference number 48, later with another number (53). I suppose that this is an error because you have added further references recommended by another reviewer.

Reviewer #2: (No Response)

7. PLOS authors have the option to publish the peer review history of their article (what does this mean?). If published, this will include your full peer review and any attached files.

Do you want your identity to be public for this peer review? For information about this choice, including consent withdrawal, please see our Privacy Policy.

Reviewer #1: No

Reviewer #2: No

---

## [Author Response · Author response to Decision Letter 1]

19 Oct 2024

Please see the attached file labeled 'Response to Reviewers'.

Thank you.

---

## [Decision Letter · Decision Letter 2]

7 Nov 2024

PONE-D-24-09289R2Coping strategies and changes of type D personality were associated with depressive tendency at 9 months after percutaneous coronary interventionPLOS ONE

Dear Dr. Izawa,

Thank you for submitting your manuscript to PLOS ONE. After careful consideration, we feel that it has merit but does not fully meet PLOS ONE’s publication criteria as it currently stands. Therefore, we invite you to submit a revised version of the manuscript that addresses the points raised during the review process.

ACADEMIC EDITOR: All issues raised by one reviewre are required >==============================

We look forward to receiving your revised manuscript.

Kind regards,

Vincenzo Lionetti, M.D., PhD

Academic Editor

PLOS ONE

Journal Requirements:

Reviewers' comments:

Reviewer's Responses to Questions

Comments to the Author

1. If the authors have adequately addressed your comments raised in a previous round of review and you feel that this manuscript is now acceptable for publication, you may indicate that here to bypass the “Comments to the Author” section, enter your conflict of interest statement in the “Confidential to Editor” section, and submit your "Accept" recommendation.

Reviewer #1: (No Response)

Reviewer #2: All comments have been addressed

2. Is the manuscript technically sound, and do the data support the conclusions?

Reviewer #1: Yes

Reviewer #2: Yes

3. Has the statistical analysis been performed appropriately and rigorously? 

Reviewer #1: Yes

Reviewer #2: Yes

4. Have the authors made all data underlying the findings in their manuscript fully available?

Reviewer #1: Yes

Reviewer #2: Yes

5. Is the manuscript presented in an intelligible fashion and written in standard English?

Reviewer #1: Yes

Reviewer #2: Yes

6. Review Comments to the Author

Reviewer #1: Most of my suggestions/concerns were addressed.

Only one aspect:

Page 7, line 130ff: The reported prevalence of depression or anxiety in patients with

131 CAD remains constant for up to 12 months [28, 29], increasing thereafter [28],

132 suggesting that other factors could have an effect on the long-term psychological well

133 being of these patients. The evaluation was therefore conducted at 9 months, when

134 patients were in a stable chronic phase of depressive tendencies.

From my point of view this is not logical. Why do you not assess after e.g. 24 months? It could be important to correlate your pre-assessment with long-term outcome with increased values for anxiety and depression.

Reviewer #2: (No Response)

7. PLOS authors have the option to publish the peer review history of their article (what does this mean?). If published, this will include your full peer review and any attached files.

Do you want your identity to be public for this peer review? For information about this choice, including consent withdrawal, please see our Privacy Policy.

Reviewer #1: No

Reviewer #2: Yes: Pasquale Bufano

---

## [Author Response · Author response to Decision Letter 2]

11 Dec 2024

Please see the attached file. Thank you for your kind reviewing this manuscript.

---

## [Decision Letter · Decision Letter 3]

16 Dec 2024

Coping strategies and changes of type D personality were associated with depressive tendency at 9 months after percutaneous coronary intervention

PONE-D-24-09289R3

Dear Dr. Izawa,

We’re pleased to inform you that your manuscript has been judged scientifically suitable for publication and will be formally accepted for publication once it meets all outstanding technical requirements.

Kind regards,

Vincenzo Lionetti, M.D., PhD

Academic Editor

PLOS ONE

Additional Editor Comments (optional):

Reviewers' comments:

Reviewer's Responses to Questions

**Comments to the Author**

1. If the authors have adequately addressed your comments raised in a previous round of review and you feel that this manuscript is now acceptable for publication, you may indicate that here to bypass the “Comments to the Author” section, enter your conflict of interest statement in the “Confidential to Editor” section, and submit your "Accept" recommendation.

Reviewer #1: All comments have been addressed

2. Is the manuscript technically sound, and do the data support the conclusions?

Reviewer #1: Yes

3. Has the statistical analysis been performed appropriately and rigorously? 

Reviewer #1: Yes

4. Have the authors made all data underlying the findings in their manuscript fully available?

Reviewer #1: Yes

5. Is the manuscript presented in an intelligible fashion and written in standard English?

Reviewer #1: Yes

6. Review Comments to the Author

Reviewer #1: (No Response)

7. PLOS authors have the option to publish the peer review history of their article (what does this mean?). If published, this will include your full peer review and any attached files.

Reviewer #1: No

---

## [Editor Report · Acceptance letter]

3 Jan 2025

PONE-D-24-09289R3 

PLOS ONE

Dear Dr. Izawa, 

I'm pleased to inform you that your manuscript has been deemed suitable for publication in PLOS ONE. Congratulations! Your manuscript is now being handed over to our production team.

Kind regards, 

on behalf of

Prof. Vincenzo Lionetti 

Academic Editor

PLOS ONE